# Children and Adolescents Dietary Habits and Lifestyle Changes during COVID-19 Lockdown in Italy

**DOI:** 10.3390/nu14102135

**Published:** 2022-05-20

**Authors:** Giovanni Farello, Marianna D’Andrea, Alessia Quarta, Armando Grossi, Davide Pompili, Emma Altobelli, Stefano Stagi, Clara Balsano

**Affiliations:** 1Department of Clinical Medicine, Public Health, Life and Environmental Sciences, University of L’Aquila, 67100 L’Aquila, Italy; mary.dandrea1988@libero.it (M.D.); alessiaquarta54@gmail.com (A.Q.); davide.pompili@gmail.it (D.P.); emma.altobelli@univaq.it (E.A.); 2Department of Pediatrics-Endocrinology, Bambino Gesù Children’s Research Hospital, 00165 Rome, Italy; arma.gro59@gmail.com; 3Department of Health Sciences, University of Florence, Anna Meyer Children’s University Hospital, 50100 Florence, Italy; stefano.stagi@unifi.it; 4Department of Life, Health & Environmental Sciences, School of Emergency and Urgency Medicine, University of L’Aquila, 67100 L’Aquila, Italy; clara.balsano@univaq.it

**Keywords:** pandemic, home-confinement, dietary habits, obesity

## Abstract

Background: We evaluated the changes in lifestyle during the COVID-19 pandemic lockdown in a sample of children and adolescents in order to assess any increase in risk factors for the onset of cardiovascular diseases in later ages. Methods: We conducted a cross-sectional study involving 965 parents who completed an online survey about dietary habits and lifestyle during the first lockdown in Italy (from 9 March 2020 to 18 May 2020) and compared their findings with the period before the pandemic. The inclusion criteria were parents (or caregivers) with Italian residency and with children aged between 5 and 18 years. Results: We identified 563 adolescents and 402 children. The mean age was 12.28 years (SD 3.754). The pandemic was associated with an increase in the consumption of high-calorie snack foods. The total amount of food in homes during lockdown compared with before the pandemic increased 50%. Relating to the parent-perceived child weight status, more parents reported obesity in their children after lockdown (+0.6% in the 5–11 age group and +0.2% in the 12–18 age group). We reported a reduction of physical activity, an increase of sedentary lifestyle and sleep habits changes. Conclusion: The COVID-19 pandemic was associated with changes in the lifestyles of children and adolescents; this could cause an increase in the incidence of obesity and of cardiovascular and metabolic diseases in adulthood.

## 1. Introduction

In December 2019, coronavirus-2 (SARS-CoV-2) emerged in Wuhan, China, transmitted from animals to humans [1]. It rapidly spread from Wuhan to the rest of the world, causing the acute respiratory syndrome called COVID-19 [2]. After that, on 30 January 2020, the World Health Organization (WHO) declared SARS-CoV-2 a global pandemic [3,4]. In Italy, on 10 March 2020, the government imposed a total lockdown at the national level to contain the virus [5]. Due to the restrictive containment measures adopted during the lockdown, children and adolescents were subjected to highly stressful situations. School closures, interruption of physical activities and home confinement determined significant changes in dietary habits, daily routines, sleep behaviours and the mental health of families [6]. Consequently, people stayed at home and started stockpiling food. In addition, the interruption of work routines led to boredom and in turn greater food intake [7,8]. Lack of structured time, such as over the summer break for schoolchildren, is often associated with obesogenic behaviours, which are much less frequent when children have scheduled routines [9]. Maria Belen Ruiz-Roso et al. [10] studied changes in adolescents’ diets in Italy, Spain, Chile, Colombia and Brazil during COVID-19 confinement and identified improved eating habits that involved increasing legume, fruit and vegetable intake, even though this, apparently, did not increase the overall diet quality. Adolescents showed greater consumption of sweets, likely due to boredom and stress produced by COVID-19 confinement. This study shows the associations between gender, country of residence, family members at home, and watching TV during mealtimes. In an adult population [11], some changes in eating habits were identified, for instance, increased consumption of sweets and other comfort foods; other lifestyle changes were related to sleep-wake rhythms and limitations in physical activities. Similar results have been documented in other studies involving adults [12,13] Von Hippel et al. (2007) demonstrated that the gain in body mass index (BMI) is greater during the summer than during the school year [14]. In a systematic review of seven studies, Franckle and colleagues (2014) [15] assessed that the accelerated weight gain in children during summer vacation was associated with decreased vegetable consumption and increased screen time [16]. During the pandemic, families tended to stock up on food to minimize social exposure [17,18]; they purchased more food, including larger amounts of less healthy nonperishable foodstuffs that were lower in macronutrients [1]. These purchasing behaviors changed the availability of food at home, influencing children’s dietary intake. Moreover, greater stress, fewer economics resources, and less access to food resulted in stricter control over feeding practices [19,20]. These have been associated with the poorer regulation of energy intake and adiposity in children [21]. The increase in food consumption during COVID-19 increased children’s risk of obesity [22] and cardiovascular risk in adulthood [23,24]. Moreover, families have lost their jobs and seen their incomes decreased because of COVID-19 impacts; in consequence, the economic impacts of the lockdown have to be taken in account [25]. In general, poverty predisposes low-income families towards suboptimal diets [26]. Children growing up in households with limited financial resources present obesity more frequently than do children growing up with more financial resources [27]. Socioeconomic disparities are a well-documented limitation on access to physical activity [28], and social distancing requirements widened the gap even more [29]. In our study, we evaluated the changes in lifestyle and eating habits during lockdown in a sample of children and adolescents. Our aim was to detect any increase in risk factors for developing cardiovascular diseases in later ages.

## 2. Subjects and Methods

The survey background information with the instructions for completing and the link to participate in the study were sent to schools. The inclusion criteria were the following: Italian citizenship, absence of chronic diseases and/or food allergies and age between 5 and 18 years. We collected data from 7 to 18 January 2021. Parents with more than one child in this age range could complete the questionnaire separately for each child. At the beginning of the survey, parents were asked to sign an informed consent form. Parents were randomly recruited using the survey link through telematics platforms. The full questionnaire took approximately 10 min to complete. In total, 972 parents completed the survey, but 2 surveys were deemed invalid because some answers were lacking, 4 were excluded because of the omission of a child’s age, and 1 respondent did not meet the inclusion criteria. Thus, the final child sample was of 965 subjects, 531 males and 434 females with a mean age of 12.28 years. Collected data were then divided into two age groups: the first one was the 5–11 years group (total: 402 valid forms), namely, the child group, and the second one was the 12–18 years group (total: 563 valid forms), namely the adolescent group. The questionnaire referred to the first lockdown in Italy, which occurred from 9 March to 18 May 2020, compared with the period before the pandemic, and it was divided into the 5 sections described below.

(1)Demographics and COVID-19 related questions.

These questions regarded child age, sex, nationality (Italian/foreign), parent education, number of children and adults in the home, parent-perceived child weight status and changes in physical activities before and after the lockdown; diagnosis of COVID-19 in at least one family member in the home; increased time spent at home during lockdown; income reduction; loss of parent job; and state economic grants.

(2)Household food environment.

Seven questions were about the amount of total food, high-calorie snack foods, desserts, sweets, high-carbohydrate foods, and non-perishable processed foods kept in homes and the amounts of fast-food meals and home-cooked meals with fresh foods during and before the pandemic.

(3)Child feeding practices.

Three questions were about the amount of total food eaten and the number of meals containing fresh foods during the lockdown.

(4)Parent feeding practices.

Parent feeding practices were assessed using 4 questions from the Child Feeding Questionnaire (CFQ) [30]. The questions included parents’ concerns about their children’s weight during the lockdown; monitoring (i.e., the extent to which parents supervised their child’s eating), restriction (i.e., the extent to which parents restrict their children’s to certain foods) and pressure to eat.

(5)Lifestyle habits.

Twenty-seven questions regarding sedentary lifestyles, physical activity and hours of sleep during lockdown. Questions about physical activity habits were modified from the “Youth in Review Activity Profile” YAP27, a questionnaire used to measure parameters in paediatric age and structured in 3 parts (physical activity at school, extra-curricular physical activity, sedentary activities) [31].

Statistical analysis: We describe samples by means and standard deviations (SD) in relation to the quantitative variables and frequencies in relation to qualitative ones. To evaluate the associations between changes in household food environment and age group, Chi-square test was used. For multiple comparisons, data were analysed with a two-factor repeated-measures analysis of variance (ANOVA). Statistical analysis was performed with SAS statistical software (version 8.12, 2000; SAS Institute Inc., Cary, NC, USA).

## 3. Results

### 3.1. Demographics and COVID-19 Related Questions

The sample (Table 1) consisted of N = 402 children (5–11 years) and N = 563 adolescents (12–18 years). Total sample mean age was 12.28 years (SD ± 3.754), with a prevalence of boys. There was a high prevalence (>95%) of children and adolescents with Italian nationality. Parents showed high education levels; 70% held a diploma or a degree. For the total number of family members in a household, 70% of parents reported more than 3 members in both age groups (children and adolescents). Concerning the parent-perceived child weight status, we found more reports post lockdown of overweight (+5.7% in the 5–11 age group and +0.8% in the 12–18 years) and obesity (+0.6% in the 5–11 age group and +0.7% in the 12–18 age group) and fewer reports of normal weight or underweight. In most (90%) of the cases, time spent at home increased. About 44% of families were affected by an income reduction, about 35% of families received a state economic grant.

### 3.2. Household Food Environment

The total amounts of food in homes during the lockdown either increased or remained unchanged, decreasing in only a few cases and with approximately with the same frequency in both age groups. Nonperishable foods increased in about 50% of both groups and decreased only in 6% of both groups. Consumption of desserts and high-carbohydrate foods during lockdown increased significantly in both groups compared with before lockdown. Fast food and take-away foods showed a decrease of 53.8% and an increase of 5.6% among the children; in the 12–18 years old age group, consumption of fast food remained unchanged at in 44.6% of the group 12–18 years old and take-away consumption increased 12.9% compared with the consumption before pandemic.

### 3.3. Child Feeding Practices

Most parents (71.2%) replied that during lockdown, their children’s food intake remained unchanged, and 25% of them reported an increase; only 3.7% reported a reduction. Among adolescents, 9% of parents reported lower food intake, 66.7% said intake was unchanged and 24.3% said their adolescents’ intake had increased. Consumption of meals containing fresh foods remained unchanged between lockdown and before the pandemic, 81%, with no differences between samples (Table 2).

### 3.4. Lifestyle Changes during COVID-19 Pandemic

There was a significant difference (*p* < 0.01) between the percentage of children and adolescents training before and during lockdown for COVID-19 (Table 3), although no significant differences were found between age groups (*p* = 0.892). Physical activity of other types (such as playing in the garden or around the neighbourhood) showed significant reductions during lockdown. Before the pandemic, only 6.6% of children never engaged in out outdoor activities, but the percentage increased to 37.8% during lockdown; some gyms and sports associations proposed online lessons, but 71.8% of children and 64% of adolescents never took part.

### 3.5. Data about Screen Time

Parents reported increases in sedentary lifestyles: The percentages of children watching TV up 2 h/day increased from 9.6% to 34.8% during the lockdown period; video game and internet use increased also in both age groups. The percentage of adolescents who spent more than 3 hours/day online increased from 6.2% to 47.8%. Significant differences emerged in both age groups between before and during the lockdown (*p* < 0.01) and between the two groups (*p* < 0.001) in screen time and video game and internet use. However, school-related videos contributed greatly to these changes in lifestyle: During lockdown, the percentage of children who performed school-related activities for 2–3 h/day or for more than 3 h/day increased from 0% to 16.7% and 14%, respectively. In adolescents, this percentage increased from 1.4% to 75.6% (*p* < 0.01) (Table 4).

## 4. Discussion

There are still few studies regarding children’s and adolescents’ lifestyle changes during lockdown. To the best of our knowledge, the unique studies conducted with Italian children have been by Pietrobelli and colleagues (2020) and Ruiz-Roso and Pujia, R [10,32,33]. The first study explored the changes in lifestyle associated with COVID-19 among 41 children and adolescents with obesity. It revealed that sports activities decreased, sleeping time increased and screen time increased significantly to about 4 h a day; the latter exceeded the recommendations provided by the American Academy of Pediatrics [34]. However, that study did not refer to information gathered before lockdown and was specifically related to children who were already in treatment for obesity. The second study [10] suffers from considerable conditionality based on the different (i.e., not Italy) country of origin. Pujia [33] found a change in eating habits with an increase in consumption of sweet packaged snacks and processed meat, as well as of bread, pizza and bakery products. A total of 59.7% of the participants reported body weight gain, with adolescents gaining weight more than children. Moore et al. (2020) conducted a survey of physical activity among 1472 Canadian children and teens between the ages of 5 and 17 years [35]. Only 4.7% of children between 5 and 11 years followed the Physical Activity Guidelines recommendations, according to which they should practice 60 min or more of moderate-to-vigorous physical activity daily [36]. This study demonstrated that the encouragement of parents and their involvement in practicing physical activity are positively associated with healthier behaviours in children.

### 4.1. Heating Habits

The current survey represents one of the first transversal studies examining the effects of the COVID-19 pandemic on eating habits in Italian families. Data collected revealed changes both in total amount of food in the home because of “panic shopping” [37,38,39] and in the types of food compared with the pre-pandemic period. As a result, there was a notable increase in consumption of non-perishable and high-carbohydrate foods, proceeding from the rational desire for families to stock up on foods and minimize social exposure. During lockdown, there was an increase in home-cooked meals since parents spent more time at home. Interestingly, Italy has ranked first in Europe for the rediscovery of home cooking with an increase in the so-called “MasterChef effect” [40,41]. Data collected also revealed a significant decline in take-away food consumption, particularly in families with children between 5 and 11, years due to the total closure. However, among adolescents, the percentage of fast-food meals remained unchanged, probably because of adolescents’ adeptness using food delivery applications.

### 4.2. Perception of Body Weight

About parent-perceived weight status of children, some interesting considerations emerged; in particular, parents were revealed to have inaccurate perceptions of their children’s weight perception; only 12% of parents accurately recognized real weight gain in their children. This could depend on the cultural level, parents’ own physical health and the social context. During lockdown, more parents of children than adolescent reported overweight and obesity, with about +5% overweight and +0.6% obesity in children against about +1% and +0.2% in adolescents. We must specify that no anthropometric measurements were taken in consequence parents could have underestimated their children’s weight. We explain these data by the fact that childhood is usually a susceptible period for the onset of excess weight due to some crucial phases: pregnancy, weaning and adiposity rebound. In contrast, most adolescents had less weight gain during the pandemic, probably due to increased physical activity levels. The somatization of family problems and stress due to social restrictions affected children’s behaviour. These factors together with the lack of sport activity combined to increase food intake. From the biological point of view, prolonged stress activates the hypothalamus-pituitary axis, which in turn induces the release of cortisol by the adrenal gland [42,43]. Cortisol has an anorectic effect because it induces catabolic processes, increasing blood sugar, lipolysis and proteolysis, but it also stimulates appetite [44,45,46]. The anorectic effect in adolescents, together with the fear of gaining weight and the increased quantity of food at home, may have caused or aggravated a conflictual relationship with food, inducing eating disorders like anorexia or bulimia [47,48,49].

### 4.3. Others Lifestyle Changes

Most of the subjects spent their time watching television, playing video games and performing school-related activities. The adolescents were deprived of their social lives, obliged to stay at home and maintain only virtual contact with their friends. One important finding was that only 6.6% of children did not practice any kind of physical activity before the pandemic but that the percentage increased to 37.8% during lockdown. In adolescents, these percentages raised from 22.4% to 59.1%. This effect depended on several factors, including the availability of an outdoor area for play and living in a city versus the countryside. Conversely, children and adolescents who practiced physical activity every day before the pandemic continued training daily during lockdown. Although most of parents were aware of increases in daily food intake and in sedentary lifestyles, most of them did not reduce their children’s access to food.

The aim of this study was to describe changes in eating habits, the home food environment, parent feeding practices, physical activities and sleeping behaviours of Italian children and adolescents during the COVID-19 lockdown in order to promote public health strategies to prevent impacts of the pandemic on children’s and adolescents’ health. 

The main limit of the study is that it is a survey, so it reflects the data perceived by parents, and not reality, furthermore being a voluntary survey, the the data collected could be biased and based solely on the perception of the parents.

This issue needs additional studies aimed better understanding how home confinement influences paediatric health status. Understanding whether a pandemic can indirectly cause or aggravate obesity will be essential for preventing long-term consequences. This study has some limitations such as the short observational period and the fact that it was a voluntary survey and that could have introduced significant bias. However, the study still it provides a reliable picture of a limited slice of Italian population.

## 5. Conclusions

This study underlined significant changes in the lifestyles of children and adolescents in Italy during the COVID-19 pandemic. These were associated with increased incidence of obesity, as well as of cardiovascular and metabolic diseases in adulthood. Thus, it is extremely important to promote preventive measures especially in children. As a suggestion, pediatricians could offer guidance to families promoting healthy eating habits and daily physical activity.

## Figures and Tables

**Table 1 nutrients-14-02135-t001:** Demographics and COVID-19 Related Questions.

	5–11 Years Old N = 402		12–18 Years Old N = 563	
Age years old mean (DS)	8.45 ± 2.02		15.02 ± 1.80	
Child sex				
Male	47.6%		60.3%	
Female	52.4%		39.7%	
Child nationality				
Italian	98.5%		97.5%	
Foreign	1.5%		2.5%	
Parents nationality	Father	Mother	Father	Mother
Italian	97.0%	95.3%	96.6%	95.5%
Foreign	3.0%	4.7%	3.4%	4.5%
Parents education	Father	Mother	Father	Mother
Primary school license	1.5%	0.2%	0.7%	0.5%
Middle school license	25.5%	10%	32.8%	19.1%
Diploma	53.8%	45.4%	49.5%	55.2%
Degree	19.2%	44.4%	16.9%	25.2%
Number of family members in the home				
≤3	27.1%		28.4%	
>3	72.9%		71.6%	
Parent-perceived child weight status	Pre-lockdown	Post-lockdown	Pre-lockdown	Post-lockdown
Underweight	6.0%	3.5%	5.2%	5.3%
Normal	82.0%	65.9%	82.7%	81.7%
Overweight	12.0%	17.7%	11.6%	12.4%
Obese	0.0%	0.6%	0.5%	0.7%
Parent-perceived physical lifestyle	Pre-lockdown	Post-lockdown	Pre-lockdown	Post-lockdown
More sedentary	2.7%	45.4%	5.5%	68.4%
Sometimes sedentary	24.7%	36.8%	38.0%	22.7%
Rarely sedentary	50.2%	13.5%	35.6%	6.5%
Never sedentary	23.1%	4.2%	20.9%	2.4%
COVID-19 diagnosis in at least one family member in the home				
No	90.9%		94.3%	
Yes	9.1%		5.7%	
Increased time spent at home by parents during lockdown				
Neither	10.8%		12.3%	
Only one	39.5%		40.4%	
Both	49.8%		47.2%	
Income reduction because of lockdown				
No	59.1%		54.4%	
Yes	40.9%		45.6%	
Lost of at least one parent’s job because of lockdown				
No	81.4%		81.8%	
Yes	18.6%		18.2%	
State economic grant				
No	59.9%		68.8%	
Yes	40.1%		31.2%	

**Table 2 nutrients-14-02135-t002:** Household food amounts during lockdown compared with before the pandemic.

	5–11 Years Old N = 402(%)	12–18 Years Old N = 563(%)	*p* Value
Total food in home			n.s
Decreased	7.7	11.8	
Unchanged	41.5	44.7	
Increased	50.7	43.5	
Nonperishable processed foods in home			n.s
Decreased	4.5	7.5	
Unchanged	42.3	43.2	
Increased	53.2	49.3	
High-calories snack foods in home			n.s
Decreased	17.7	20.5	
Unchanged	56.2	53.4	
Increased	26.1	26.1	
Sweets and desserts in home			* vs ** *p* < 0.05
Decreased	6.0 *	11.6 **	
Unchanged	45.5	43.9	
Increased	48.5	44.6	
High-carbohydrate foods in home			* vs ** *p* < 0.05
Decreased	1.7 *	4.6 **	
Unchanged	44.8 *	50.7 **	
Increased	54.0	44.7	
Take-out/fast food/already prepared meals			* vs ** *p* < 0.05
Decreased	53.8 *	42.4 **	
Unchanged	40.6	44.6	
Increased	5.6 *	12.9 **	
Home-cooked meals with fresh foods			n.s
Decreased	7.2	5.7	
Unchanged	39.7	42.5	
Increased	53.1	51.8	

n.s. = not significant; * vs ** statistical significance between values.

**Table 3 nutrients-14-02135-t003:** Training before and during the COVID-19 pandemic.

	5–11 Years Old N = 402(%)		12–18 Years Old N = 563 (%)		*p*^1^ Value *	*p*^2^ Value°
Training/week	PRE	DURING COVID-19	PRE	DURING COVID-19		
No training	11.7	87.6	12.3	80.1		
1 times/week	9.8	4.0	4.1	6.9	<0.001	n.s
2 times/week	50.5	6.4	27.4	6.5		
3 times/week	24.5	1.3	47.3	2.7		
Daily	3.6	0.8	8.9	3.8		
Free play, going for a	PRE	DURING-COVID-19	PRE	DURING COVID-19		
walk in their neighborhood						
Never	6.6	37.8	22.4	59.1	<0.001	n.s
1 times/week	13.2	15.3	21.0	12.0		
2 times/week	25.1	10.6	24.1	10.3		
3 times/week	23.1	11.3	14.5	6.2		
Daily	31.9	25.0	17.9	12.4		

n.s. = not significant, * *p*^1^ reflects the effect of time between the values before COVID-19 and during the COVID-19 pandemic in each group; **°**
*p*^2^ reflects the effect of either the 5–11 group or the 12–18 group on physical activity before and during the COVID-19 pandemic.

**Table 4 nutrients-14-02135-t004:** Inactive lifestyle.

	5–11 Years Old N = 402(%)			12–18 Years Old N = 563 (%)	*p*^1^ Value *	*p*^2^ Value °
TV hours/day	PRE	DURING COVID-19	PRE	DURING COVID-19		
Never	3.8	1.5	14.1	12.8		
1 h	42.3	11.4	35.4	14.1	<0.001	<0.001
1–2 h	42.1	21.8	37.8	22.8		
2–3 h	9.6	34.8	10.3	25.5		
>3 h	2.3	30.5	2.4	24.8		
Videogames/Internet/day	PRE	DURING-COVID-19	PRE	DURING COVID-19		
Never	37.9	17.2	4.8	2.1	<0.001	<0.001
1 h	43.4	24.6	35.1	8.2		
1–2 h	13.4	23.1	40.2	13.7		
2–3 h	3.8	18.7	13.7	28.2		
>3 h	1.5	16.4	6.2	47.8		
School related video	PRE	DURING COVID-19	PRE	DURING COVID-19		
activities						
Never	86.8	15.9	45.2	0.3	<0.001	n.s.
1 h	8.9	33.0	38.3	3.1		
1–2 h	4.3	20.5	9.0	5.8		
2–3 h	0.0	16.7	6.2	15.1		
>3 h	0.0	14.0	1.4	75.6		

n.s. = not significant, * *p*^1^ reflects the effect of time between before and during the COVID-19 pandemic in each group, **°**
*p*^2^ reflects the effect of either age 5–11 or age 12–18 on inactive lifestyle before and during the pandemic.

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
