# Peer review of "Children and Adolescents Dietary Habits and Lifestyle Changes during COVID-19 Lockdown in Italy"

_nutrients, 2022, doi:10.3390/nu14102135_

Round 1

Reviewer 1 Report

The manuscript submitted to Nutrients by Farello et al., titled: "Children and adolescents dietary habits and lifestyle changes during COVID 19 lockdown in Italy" is human work investigating the dietary habits and lifestyle changes related during the COVID-19 lockdown period in Italy. The manuscript covers an interesting and important topic that has been on the scientific debate as per nutritional-health implications and side-effects of the pandemic.

The manuscript is well structured and organized logically while it is easy for the reader to follow.

The reviewer would like to make the following points for the authors' consideration:

  1. Consider outlining the inclusion and exclusion criteria.
  2. Was the questionnaire used validated?
  3. What were the confounding factors identified/taken into consideration? (e.g.: socioeconomic status, education, living conditions, nutrition knowledge differences etc).
  4. What was the rationale for the mode of selection and the number of participants? Was there a power calculation? How did the authors safeguarded against potential confounders in terms of the area of residence?
  5. How do results of this study compare to similar studies in other settings? Is the Italian setting different in any way than other settings?
  6. The manuscript is weak in terms of addressing the current literature. Even though COVID is a new situation there is already a fair amount of literature considering the change of dietary habits in various countries/settings and varying ages including those the authors focus on. Here are some manuscripts to consider even though not all of them cover the same age range still there is valuable information and aspects to be discussed:
    1. Sidor A, Rzymski P. Dietary Choices and Habits during COVID-19 Lockdown: Experience from Poland. Nutrients. 2020 Jun 3;12(6):1657. doi: 10.3390/nu12061657. 
    2. Martínez-de-Quel Ó, Suárez-Iglesias D, López-Flores M, Pérez CA. Physical activity, dietary habits and sleep quality before and during COVID-19 lockdown: A longitudinal study. Appetite. 2021 Mar 1;158:105019. doi: 10.1016/j.appet.2020.105019.  
    3. Ruiz-Roso MB, de Carvalho Padilha P, Mantilla-Escalante DC, Ulloa N, Brun P, Acevedo-Correa D, Arantes Ferreira Peres W, Martorell M, Aires MT, de Oliveira Cardoso L, Carrasco-Marín F, Paternina-Sierra K, Rodriguez-Meza JE, Montero PM, Bernabè G, Pauletto A, Taci X, Visioli F, Dávalos A. Covid-19 Confinement and Changes of Adolescent's Dietary Trends in Italy, Spain, Chile, Colombia and Brazil. Nutrients. 2020 Jun 17;12(6):1807. doi: 10.3390/nu12061807. 
    4. Husain W, Ashkanani F. Does COVID-19 change dietary habits and lifestyle behaviours in Kuwait: a community-based cross-sectional study. Environ Health Prev Med. 2020 Oct 12;25(1):61. doi: 10.1186/s12199-020-00901-5.
  7. The manuscript would benefit from a better organization of the discussion in paragraphs conceptually as opposed to a continuous narrative.
  8. Proofreading for improvement of text (flow and expressions) as well as correcting typos is recommended.

Author Response

Reviever 1

Dear Reviewer,

Thank you for the attention you have placed in reading our study and for the very correct observations you have made that will allow us to improve the study.

1) in the material and methods section we have added the exclusion criteria

2) the questionnaire relating to the evaluation of eating habits was validated by the authors cited in the bibliography:

Birch LL, Fisher JO, Grimm-Thomas K, Markey CN, Sawyer R, Johnson SL. Confirmatory factor analysis of the Child Feeding Questionnaire: a measure of parental attitudes, beliefs and practices about child feeding and obesity proneness. Appetite. 2001 Jun; 36 (3): 201-10. doi: 10.1006 / appe.2001.0398. PMID: 11358344.

3) The sample under study was homogeneous as it came from regions of Central Italy alone, with the same level of school education.

4) The study was carried out in the Italian region of Abruzzo, whose population is 1200,000 inhabitants, calculating a margin of error of 5% and a confidence interval of 95% require at least 385 responses.

5) On the basis of the present literature, the results do not seem to be discordant with those obtained in other settings

6) In the correct manuscript we have reported the literature you suggested and we have taken into account that some of the works had a series of adult subjects.

7) We corrected the discussion by dividing it into the main paragraphs

Thank you

Reviewer 2 Report

The subject of the study entitled "Children and adolescents dietary habits and lifestyle changes during COVID 19 lockdown in Italy", is very interesting and could inform about the impact of confinement on children and adolescents.

But I believe that there are important limitations that need to be clarified in the methods.

2.Subjects and Methods

“The current cross-sectional and observational study used an online survey developed by Google.”

The way of knowing children's habits and their changes through parent surveys should be justified and clarified. In addition, an online survey developed by Google is used that is not justified, nor is it explained to whom it was offered. (if it is free for an entire school, neighborhood or parents in general). How do you access the survey?

The article does not say how the participants access the questionnaire. How the participants are recruited, or how they are told that they have the questionnaire open to carry it out.

The response rate to the questionnaire cannot be known and there may be a clear response bias. Results may not be representative of the population. There is also no analysis of whether the addresses of the study participants are representative of the population, economic level, education...

It is not indicated how the questionnaire has been validated

“the lack of information about the participant social status and the low sample number make it difficult to generalize the results”

This sentence is not enough

in the affirmation

"However, among adolescents, the percentage of fast-food meals remained unchanged, probably because of the adolescent's practicality in use of food delivery applications."

If all the food delivery restaurants were closed, would the teenagers not be able to shop?

In addition, this information is collected by the survey carried out by parents.

In short, all these limitations of the study should be included and explained in the article in the methods and before the conclusions.

Author Response

Dear Reviewer,

Thank you for the attention you have placed in reading our study and for the very correct observations, you have made which will allow us to improve the study.

1) The rationale of the study, the explanations for the compilation, the inclusion / exclusion criteria and the informed consent were sent by email to the parents through the school institution. We hope we have clarified this by correcting the materials and methods section.

2) The study was addressed to a relatively homogeneous population residing in the same province; the minimum number of 385 responses was calculated based on a total population of 1200000 inhabitants with an error margin of 5% and a confidence interval of 95%. The answers we obtained were from 563 adolescents and 402 children.

3) The questionnaire, relative to eating habits, was the CFQ validated by the authors: Birch, L. L. et al. Confirmatory factor analysis of the Child Feeding Questionnaire: A measure of parental attitudes, beliefs and practices about child feeding and obesity proneness. Appetite 36, 201–210 (2001).

4) We eliminated the phrase "Moreover, the lack of information about the participant social status and the low sample number make it difficult to generalize the results." And we have limited ourselves to reporting the results of the study.

Reviewer 3 Report

REVIEW REPORT: Children and adolescents dietary habits and lifestyle changes during COVID 19 lockdown in Italy

The present article evaluates the changes in lifestyle during pandemic Covid-19 lockdown in a sample of Italian children and adolescents in order to assess any increase in risk factors for the onset of cardiovascular diseases in later ages.

The article is written in an appropriate way, the objectives are defined and the results and data are presented in an appropriate way. The main weakness of the study is that it is a survey, so it only reflects the data perceived by the parents, and not reality. In addition, there are many studies of this type published recently, so the content does not provide new knowledge.

 Specific comments:

Introduction:

The introduction provides sufficient background and include relevant references.

The aims of the work are clearly defined at the end of the introduction.

Subjects and Methods:

The intervention is well described. The only point I don’t understand very well is how the parents were randomly recruited using the survey link through telematics platforms. It would also be good to clarify if the survey is a self-completed parent questionnaire.

There are not exclusion criteria described, so it means that every parent was eligible, correct?

 Results:

 The results are presented appropriately.

 Table 1 should be review because the lines are moved.

 Pay attention to the phrase: “in 12-18 y age group consumption of takeaway foods remained unchanged in 44.6% or increased in 44.7% compared to before pandemic.” According to table 2, this last percentage remains to high-carbohydrate foods in home, nor to takeaway foods. It also be corrected the word carbohidrate in the table (correct: carbohydrate).

            “The amount of meals containing fresh foods remained unchanged in the lock- down and before the pandemic (81%),”: this statement does not agree with the results presented in table 2, review.

            When talking about lifestyle changes during COVID-19 pandemic, the results presented in table 3 should not be repeated in the text so as not to be redundant. Discuss only the most relevant aspects. The authors also may be careful with the table alignments, they are moved.

Discussion:

Authors say that the unique study conducted on Italian children is from Pietrobelli and colleagues in 2020, but there is another one:

Pujia R, Ferro Y, Maurotti S, Khoory J, Gazzaruso C, Pujia A, Montalcini T, Mazza E. The Effects of COVID-19 on the Eating Habits of Children and Adolescents in Italy: A Pilot Survey Study. Nutrients. 2021 Jul 30;13(8):2641. doi: 10.3390/nu13082641. PMID: 34444801; PMCID: PMC8400531.

Regarding to the phrase: “During lockdown, overweight and obesity were more prevalent in children than in adolescents,…” The study reflects the parent-perceived child weight status, but it doesn’t have any anthropometric measure, so I think it should be clear when talking about this. Parents' perception of their children's weight may underestimate reality, so these percentages are probably not real. The study reflects only the results of a survey, and this should be made clear when discussing the results.

Possible limitations of the study are included at the end of the paper, but I think it should be added that, as it is a voluntary completion survey, the data collected could be biased.

References:

The references are appropriate and there are not self-citations. References 20, 28 and 36 must be reviewed due to the lack of journal name.

Author Response

Dear Reviewer,

Thank you for the attention you have placed in reading our study and for the very correct observations you have made which will allow us to improve the study.

1) The rationale of the study, the explanations for the compilation, the inclusion / exclusion criteria and the informed consent were sent by email to the parents through the school institution.

The link on the telematic platform (Google) was sent together with the informative material. We hope to have clarified it by correcting the materials and methods section.

2) Yes is a self-completed parent questionnaire, the only exclusion criteria were the presence of chronic diseases or food allergies.

3) we have corrected the results of table 2, Thanks

4) We followed your suggestion by eliminating the redundant part of Table 3

5)We corrected the discussion based on your observation

Round 2

Reviewer 1 Report

The authors have made a reasonable effort to address reviewer's points. Proofreading is recommended.

Reviewer 2 Report

Dear authors, your responses and changes made to the article are appropriate. Thank you

Reviewer 3 Report

Corrections have been made appropriately. Although I think the study is not very original, it is adequately written.